# The Role of Amylose in Gel Forming of Rice Flour

**DOI:** 10.3390/foods12061210

**Published:** 2023-03-13

**Authors:** Jinmu Tian, Likang Qin, Xuefeng Zeng, Pingzhen Ge, Jin Fan, Yong Zhu

**Affiliations:** 1School of Liquor and Food Engineering, Guizhou University, Guiyang 550025, China; 2Bijie Institute of Agricultural Sciences, Bijie 551700, China

**Keywords:** amylose, gel, rice flour, viscoelasticity

## Abstract

In this study, Glutinous rice (GR), Japonica rice (JR), and Indica rice (IR), with amylose contents at 1.57 ± 0.18%, 15.88 ± 1.16%, and 26.14 ± 0.25%, respectively, were selected to reveal the role of amylose in the gel forming of rice flours. The strength and elasticity of the associated gels were found in an ascendant order with the increase in amylose content. For the retrograded gels (at 4 °C for 7 days), the peak temperature (T_r_p) was positively related to the amylose content. In general, T_r_p of IR increased to 63.21 ± 0.13 °C, and the relative crystallinities of IR were in the top ranking at 10.67 ± 0.16%, followed by those of JR and GR. The relative amounts of short-range ordered structures to amorphous regions in JR and IR were also higher than that of GR, and the number of compact network structure were positively related to the amylose content. These results indicated that amylose can enhance the strength and elasticity of gels by facilitating the formation of crystalline, short-range ordered, and compact network structures. These results can provide a reference for the development of rice products.

## 1. Introduction

Rice is one type of staple food that provides nutritional components and calories to humans. Starch, composed of amylose and amylopectin, is the dominant constituent of rice [1]. Amylose is composed of D-glucopyranose units linked by α-1,4-glucosidic bonds (1-4), while amylopectin is composed of D-glucopyranose units linked by α α-1,4-glucosidic bonds and α-1,6-glucosidic bonds [2]. Based on the ratio of amylose to amylopectin, rice can be classified into Glutinous rice (GR), Japonica rice (JR), and Indica rice (IR), with low, medium, and high ratios of amylose to amylopectin, respectively, in terms of producing the corresponding rice products [3].

The process of gel forming is crucial for the quality of starch products. During gel forming, starch molecules are rearranged with the aid of intramolecular and intermolecular hydrogen bonds, endowing viscoelasticity and strength to the gels [4], and the gel matrixes play an indispensable role in rice products, especially for rice noodles. Previous studies suggested that amylose molecules presented a lower size and steric hindrance than amylopectin molecules, which endowed a high fluidity for amylose molecules during gel forming [5]. In gelatinization, amylose molecules initially seeped from the starch granules and were subsequently twisted around each other to form a network [6]. In general, amylose molecules are the dominant components to regulate the process of gel forming.

Previous studies reported that the gel prepared from the sample with elevated amylose contents exhibited a compact microstructure. While the amylopectin was found to form a loose network structure [7], the amylose was prone to forming a stable crystal structure, which required a higher temperature to melt [8]. Dun et al. suggested that the strength and crystallinity of the gel prepared from rice starch were positively related to the amylose content [9], and the crystal structure may positively affect the strength of the gel. Ambigaipalan et al. suggested that the short-range ordered structures were in favor of the organization of gel structures [10].

In the present study, the gel forming process is accompanied by the reconstitution of internal structures. The strength and viscoelasticity of the gels prepared from three varieties of rice flours with different amylose contents were evaluated by a rheometer, and the role of amylose in gel forming was revealed by the variation in crystal structures, functional groups, short-range ordered structures, and microstructures.

## 2. Materials and Methods

### 2.1. Materials

Japonica rice (JR) and Indica rice (IR) were purchased from Guizhou Jinchen Agricultural Products Development Co., Ltd. (Guizhou, China), and Glutinous rice (GR) was purchased from Shanghai Saiwengfu Agricultural Development Co., Ltd. (Shanghai, China). Amylose (source: potato starch) was purchased from Sigma-Aldrich Ltd. (St. Louis, MO, USA). Amylopectin (source: maize starch) and KBr were purchased from Macklin Co., Ltd. (Shanghai, China). NaOH was purchased from Sichuan Xilong Science Co., Ltd. (Chengdu, Sichuan, China). Ethanol and acetic acid were purchased from Tianjin Fuyu Fine Chemical Co., Ltd. (Tianjin, China). Iodine was purchased from Tianjin Kemio Chemical Reagent Co., Ltd. (Tianjin, China). Potassium iodine was purchased from Guangdong Chemical Reagent Engineering Technology Research and Development Center (Shantou, Guangdong, China).

### 2.2. Determination of Amylose Content

The amylose content of rice was determined according to the National Food Safety Standards of China [11]. A series of mixed standard solutions composed of amylose and amylopectin were prepared, with the total starch contents consistently at 0.9 mg/mL and the amylose contents between 0.10 and 0.35 mg/mL. The rice sample was sequentially subjected to crushing and defatting processes and was subsequently dissolved in a sodium hydroxide solution with a concentration of 1 mg/mL. Five milliliters of each standard solution were mixed with 1 mL of acetic acid, respectively. Aliquots (2 mL) of iodine reagent (2 g of potassium iodide mixed with 0.2 g of iodine were dissolved in water, with a reconstituted volume of 100 mL) were added into these mixtures and reconstituted to 100 mL with deionized water. The absorbances of these mixtures were measured at 720 nm, and the rice samples were also subjected to the procedures of coloration reaction and absorbance determination. The standard curve was plotted for the amylose content of standard solutions and the corresponding absorbance to calculate the amylose content in the rice sample.

### 2.3. Preparation of Rice Samples

Rice was ground into powder using a grinder (JR-200, Dongguan Xinmeihua Electromechanical Equipment Co., Ltd., Dongguan, Guangdong, China), and was subsequently passed through a 100-mesh sieve, where the undersize powder was mixed with distilled water to prepare the suspension with a rice flour content of 12% (*w*/*w*). Then, the rice flour suspension was heated in a boiling water bath for 30 min to form a gel which was subsequently cooled to room temperature. One aliquot of the gel sample was subjected to rheological analysis and the other was retrograded at 4 °C for 7 days for a series of testing by X-Ray Diffraction (XRD), Fourier Transform Infrared spectrometer (FTIR), and Scanning Electron Microscope (SEM). All rice varieties were subjected to the procedures in triplicate.

### 2.4. Monitoring of Thermal Properties

The thermal properties were monitored using Differential Scanning Calorimeter (DSC) (Q2000, TA Instruments, Inc., Newcastle, DE, USA). The sealed crucible containing rice flour suspension (6–15 mg), as can be referred to in Section 2.3, was placed in the heating furnace, and the heat flow of the sample was monitored with the heating program initiating from 30 to 95 °C at a rate of 10 °C/min. The curve that indicates the evolution of heat flow with the increase of heating temperature was obtained. The onset temperature (To), peak temperature (Tp), conclusion temperature (Tc), and gelatinization enthalpy (ΔH) were computed using the TA Universal Analysis software (TA Instruments, Inc., Newcastle, DE, USA).

The sample subjected to the former heating process was retrograded at 4 °C for 7 days, and it was then heated according to the former heating program. The heat absorption curve was obtained to demonstrate the evolution of heat flow with the increase in temperature. The retrogradation enthalpy (ΔH_r_), onset temperature (T_rO_), peak temperature (T_r_p), and conclusion temperature (T_r_c) during the melting process were also computed using the TA Universal Analysis software, and the retrogradation degree (RD) was calculated by dividing ΔHr by ΔH and multiplying it by 100% [12]. All samples were subjected to the monitoring procedures in triplicate.

### 2.5. Dynamic Rheology Analysis

The gelatinized samples described in Section 2.3 were placed between a pair of parallel plates with a diameter of 20 mm, and the gap between the plates was set at 1 mm. The evolutions of storage modulus (G′), loss modulus (G″), loss tangent (Tan δ), and dynamic viscosity (η*) in the angular frequencies between 0.63 and 66 rad/s were monitored using a rheometer (Haake Mars 60, Thermo Fisher Scientific, Waltham, MA, USA) at a strain of 1% [13].

A dynamic viscosity power function was introduced to parameterize the viscoelasticity of the sample [14], as given by Equation (1).
η* *= K** ω *^n*^*^−1^(1)
where η* indicates the dynamic viscosity, *K** indicates the consistency coefficient, ˙ω indicates the scanning frequency, and *n** indicates the flow behavior index. The *K** and *n** were calculated by result fitting with Equation (1).

### 2.6. Identification of Crystal Structures

The retrograded samples described in Section 2.3 were subjected to freeze-drying and crushing processes, and were eventually filtrated by a 100-mesh sieve. The crystal structures of the undersize sample were identified using the XRD (Empyrean, PANalytical B.V., Enschede, The Netherlands) with a scanning rate of 4°/min between 5° and 40° at a diffraction angle of 2-theta (2θ), and the flours of all rice varieties were also processed following the above procedures [15]. The relative crystallinity of the sample was calculated according to Equation (2), in which the area was measured using the Origin 2018 software (Origin Lab Corporation, Northampton, MA, USA).
Relative crystallinity (%) = 100% × Crystalline area/(Crystalline area + Amorphous area)(2)

### 2.7. Determination of Functional Groups and Short-Range Ordered Structures

The functional groups and short-range ordered structures of the retrograded samples described in Section 2.3 were determined by the FTIR (Frontier, PerkinElmer, Inc., Wesley, MA, USA). The samples were dehydrated by a freeze dryer (SCIENT-18N, Ningbo Xinzhi Biotechnology Co., Ltd., Ningbo, Zhejiang, China), and were subsequently crushed into powder. The powder was filtrated by a 100-mesh sieve before grinding with KBr with the ratio of KBr to undersize at 100:1 (*w*/*w*). Then, the mixture was pressed to form a slice to be tested by the FTIR with wavenumbers between 4000 and 400 cm^−1^ at a scan number of 32, and the air was adopted as the background. The spectra in the range between 1200 and 800 cm^−1^ were deconvoluted with a half-width of 19 cm^−1^ and resolution enhancement factor at 1.9 (OMINC 8.2, Waltham, MA, USA), the peak intensities at 1047 and 1022 cm^−1^ were computed from the spectra [16]. All samples were subjected to these procedures in triplicate.

### 2.8. Monitoring of Microstructures

The microstructures of the retrograded samples described in Section 2.3 were monitored using the SEM (S-3400N, HITACHI, Japan). The samples were dried using a freeze dryer, and were subsequently segmented to obtain aliquots with dimensions equivalent to 0.5 cm × 0.5 cm × 0.3 cm. These aliquots were gilded in a vacuum evaporator prior to monitoring under an accelerating voltage of 20 kV, and the images were magnified by 1000 and 3000 times, respectively.

### 2.9. Statistical Analysis

The results were presented in the form of means ± standard deviation (*n* = 3), and the differences between each group were determined using Duncan’s test (SPSS 20.0 software), and the significant differences were demonstrated at the level of *p <* 0.05.

## 3. Results and Discussion

### 3.1. Amylose Content of Rice

The amylose contents of GR, JR, and IR were 1.57 ± 0.18%, 15.88 ± 1.16%, and 26.14 ± 0.25%, respectively. These results were similar to those from the previous study, which showed that the amylose contents of GR starch, JR starch, and IR starch were 1.65%, 15.80%, and 27.91%, respectively [9].

### 3.2. Thermal Properties of Rice Flour

Starch gelatinization consists of a series of processes, in which the starch granules occur with the sequential changes of water absorbing, expanding, and splitting, resulting in the breaking of hydrogen bonds with the opening of double helix structures, the disappearing of crystalline regions, and the formatting of a paste [17]. In this study, the thermal properties of three varieties of rice flours with different amylose contents during gelatinization were revealed by the heat flow diagram plotted according to the evolution of heat flow with the increasing heating temperature. The onset (To), peak (Tp), conclusion (Tc) temperatures, and enthalpy (ΔH) indicating the energy required to destroy the starch granules to form a paste were adopted in the analysis. In general, To and Tp showed an increasing trend to 73.71 ± 0.24 °C and 77.35 ± 0.04 °C (for IR), respectively, with the increase of amylose content, while Tc showed no obvious trend with the variation of amylose content and was between 80.41 ± 0.28 °C and 83.99 ± 0.62 °C (Table 1). The previous study also showed that Tp of rice was higher under high amylose content than under low amylose content [18]. In addition, the previous study suggested that the short chains (degrees of polymerization DP < 10) of amylopectin may negatively affect the stability of the double helix, which gave rise to the reduction of gelatinization temperatures [19]. The other study suggested that the gelatinization temperature of starch showed an increasing trend with the increase of amylose content, due to the limiting effect of amylose on the hydration of the amorphous region [20]. The gelatinization temperature range (ΔT) was calculated by subtracting To by Tc to evaluate the homogeneity of the double helix structures. The unstable double helix structures were disrupted at lower temperatures, and more stable double helix structures required higher temperatures to disrupt [21]. In the present study, the ΔT values of IR were significantly lower (*p <* 0.05) than those of GR and JR (Table 1), which indicated that the double helix structures in IR were more homogenous than those in GR and JR. The amylose content also affected ΔH of the rice flour, demonstrating a decreasing trend with the increase of amylose content. Specifically, ΔH of IR was significantly lower (*p <* 0.05) than that of GR and JR (Table 1), which was similar to the results reported by Correa et al. [22]. This phenomenon is associated with the structures of amylopectin and amylose, in which amylopectin plays a dominant role in the double helix structure whereas amylose disrupts the orderly arrangement of the starch crystalline structures [17]. Thus, starch with a higher amylose content demonstrates fewer crystalline regions, which requires lower energy to disrupt its structure in the gelatinization process.

### 3.3. Rheological Properties of Gels

Strength and viscoelasticity are the dominant indicators of rheological properties, and they can be confirmed by the magnitude of energy stored and lost in the gels under deformation (G′ and G, respectively). In general, G′ positively correlates with the strength and elasticity of the gels whereas G″ positively correlates with the viscosity of the gels [13]. For all gel samples, the ranking of G′ values was in accordance with the ranking of amylose content at the coincident scanning frequencies between 0.63 and 66 rad/s (Figure 1A). This is consistent with the previous study, suggesting that the G′ values of the gel positively correlate with the amylose content [23]. The previous study proposed that amylose possessed a smaller spatial barrier than amylopectin, which can accelerate the reconstitution of a three-dimensional gel network with excellent elasticity [24]. The G″ values of IR were similar to those of JR, and they were 4 times higher than those of GR at the coincident scanning frequencies from 0.63 to 66 rad/s (Figure 1B). A similar phenomenon was found in the previous study of Li et al., which suggested that rice with an elevated amylose content showed a higher viscosity at specific frequencies than glutinous rice [4].

Here, tan δ indicates the relative contributions of elastic and viscous properties to the viscoelasticity of the material, and was calculated by dividing G″ by G′ [4]. In the present study, the sequences of tan δ were contrary to the order of amylose content at coincident frequencies within the scanning range, and tan δ (ω = 10 rad/s) of the gel prepared from IR was one-sixth of that of GR. This suggested that the contribution of elastic to the viscoelasticity of gel increased with the increase of amylose content (Figure 1C and Table 1), which is consistent with the results of the previous study [23].

In this work, η* was also monitored to indicate the evolution of viscosity with the increase in the scanning frequency. The individual η* was computed by dividing G″-G′i by ω (where i indicates the imaginary unit and ω indicates the corresponding scanning frequency), and the curves were shown in Figure 1D. In general, η* showed an increasing trend with the increase of amylose content. The *K** value positively correlates with the elasticity of the gel, while the *n** value positively correlates with the shear thinning behavior [4]. In the present study, the *K** values of the gels prepared from IR and JR were 24.10 and 17.11 times higher than that of the gel prepared from GR, respectively (Table 1). The *n** values of the gels were between 0.0589 ± 0.0029 and 0.3844 ± 0.0029, in which *n** of the gels prepared from IR and JR were significantly lower (*p* < 0.05) than that of the gel prepared from GR (Table 1). These suggested that the gel prepared from rice flour with higher amylose content presented a higher elasticity than that prepared from GR.

### 3.4. Thermal Properties of Retrograded Gels

In the retrograding process, the broken hydrogen bonds were reconstituted to form double helix and crystalline structures [25] that are associated with the stability of the retrograded gels. In the present study, the thermal properties of the retrograded gels were used to evaluate their stability. Specifically, T_rO,_ T_r_p, and T_r_c indicate the corresponding temperatures of phase change during the melting process, and are affected by double helix structures [26]. The retrogradation enthalpy (ΔHr) indicates the energy required to disrupt crystalline structures [17]. In general, T_rO,_ T_r_p, and T_r_c showed an upward trend with the increase of amylose content, indicating that amylose can improve the stability of double helix structures (Table 2). This may suggest that amylose is in favor of the reconstitution of hydrogen bonds. ΔHr of JR and IR was significantly higher (*p <* 0.05) than that of GR (Table 2). The previous study indicated that amylose was weighted for recrystallization [9], however, in the JR and IR samples, the order of ΔHr was contrary to that of the amylose content. Li et al. suggested that the amylopectin with long chains was more prone to recrystallization than its counterpart by forming a stable double helix structure [21]. This may indicate that JR presents more amylopectin with long chains than IR. The contribution of the retrograded starch to the total gelatinized starch was also evaluated using RD, which is in accordance with the ratio of retrogradation enthalpy to gelatinization enthalpy. Overall, RD showed an increasing trend with the increase of amylose content, in which RD of IR increased to 25.79 ± 0.16% (Table 2). The previous study suggested that starch with higher amylose content showed a faster retrogradation rate and a denser starch matrix network, which facilitated starch retrogradation [27].

### 3.5. Crystal Structures of Retrograded Gels

In the retrograding process, the crystal structures are formed from the rearrangement of starch molecules. In the present study, the crystal structures of the rice flour and retrograded gels were analyzed by the XRD, and they were confirmed according to the peak positions of the XRD patterns. The crystal structures of the rice flour prepared from GR, JR, and IR were confirmed as A-type, of which the XRD patterns showed four distinct peaks near 15°, 17°, 18°, and 23°, respectively (Figure 2A). The crystal structures of the gels retrograded at 4 °C for 7 days were divergent compared with those of the corresponding rice flour, and both the gels prepared from JR and IR were confirmed as B+V-type, showing peaks near 17° (B-type) and 20° (V-type) (Figure 2B). For the retrograded gel, the B-type crystal is recomposed from amylopectin during the retrograding process, whereas the V-type crystal indicates the complex composed of the crystal recomposed from amylose during the retrograding process and the fatty acids or phospholipids [28]. The previous study suggested that the total relative crystallinity of the gel prepared from rice starch with elevated amylose content was higher than that of the counterpart prepared from GR [9]. However, in JR and IR, the relative crystallinity of the B-type crystal also increased with the increase of amylose content, whereas the V-type crystal showed no significant difference (Table 2). This may indicate that amylose can promote the recrystallization of amylopectin.

### 3.6. Functional Groups and Short-Range Ordered Structures of Retrograded Gels

The functional groups and short-range ordered structures of the gels retrograded at 4 °C for 7 days were identified using the FTIR, and the spectra are shown in Figure 2C. The absorption peaks at 925, 846, and 756 cm^−1^ in the infrared spectra corresponded to α-1,6-D-glycosidic, α-configuration, and α-1,4-D-glycosidic bonds, respectively [16]. The absorption peaks at 1160, 1078, and 1022 cm^−1^ in these spectra were attributed to the stretching vibration of C-O in the anhydrous glucose unit [29], indicating the existence of C-O in the gels. In these spectra, the absorption peak at 2932 cm^−1^ indicates the stretching vibration C-H [30], and the absorption peak at 3400 cm^−1^ indicates the stretching vibration of O-H [31], of which the transmittances showed a decreasing trend with the increase of amylose content, indicating that amyloses may accelerate the reforming of hydrogen bonds for the rice gels.

The FTIR spectra were deconvolved in the wavenumber range between 1200 and 800 cm^−1^ to obtain the corresponding absorbance chromatograms (Figure 2D) for determining the relative amount of short-range ordered structures to amorphous regions, as calculated by dividing the peak intensity at 1047 cm^−1^ by the peak intensity at 1022 cm^−1^ [32]. The relative amount of the gels prepared from JR and IR showed no significant difference, and both were significantly higher (*p <* 0.05) than that of GR (Figure 2E). This indicated that amylose can facilitate the formation of short-range ordered structures during the retrograding process.

### 3.7. Microstructures of Retrograded Gels

The microstructures of the gels retrograded at 4 °C for 7 days were monitored using the SEM (Figure 3). The gel prepared from IR was composed of compact network structures, whereas loose network structures were found in the gels prepared from GR and JR. The number of compact network structures positively correlated with the amylose content, which may be attributed to the presence of amylose molecules that were prone to reconstituting the hydrogen bonds and increasing the denseness of the starch gels [33].

## 4. Conclusions

In the process of gel forming, starch granules are initially subjected to gelatinization with the sequential changes of water absorbing, expanding, and splitting, resulting in the breaking of hydrogen bonds and the exuding of amylose. The retrograding process subsequently occurs as the crystalline and double helical structures are formed with the aid of hydrogen bonds. The amylose is prone to retrogradation due to the lower molecular size and steric hindrance, and is in favor of the strength and elasticity of gels. In this work, the gel prepared from rice flour with amylose content at 26.14 ± 0.25% shows excellent strength and viscoelasticity, resulting from the crystalline, short-range ordered, and compact network structures. These results can provide a reference for the development of rice products.

## Figures and Tables

**Figure 1 foods-12-01210-f001:**
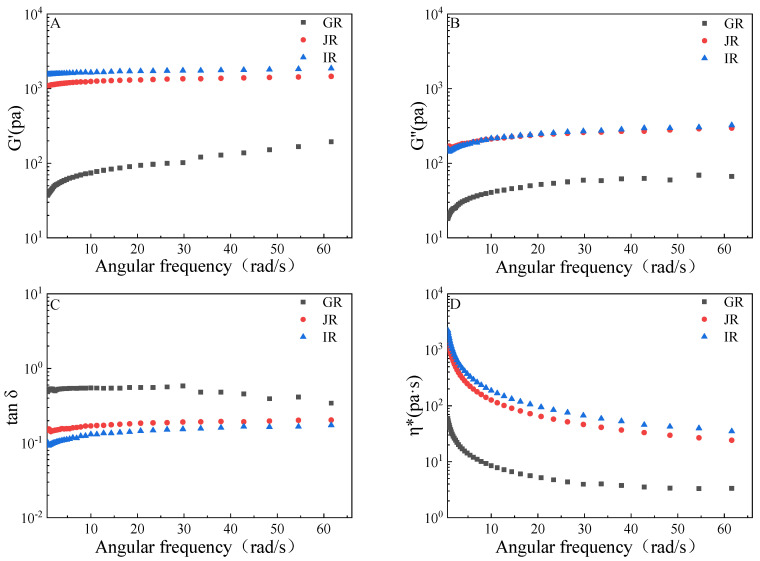
Rheological properties of the gel. (**A**–**D**) indicate the variations of the energy storage modulus (G′), loss modulus (G″), loss tangent (tan δ), dynamic viscosity(η*) with the increase of angular frequencies. GR indicates the Glutinous rice, JR indicates the Japonica rice, and IR indicates the Indica rice.

**Figure 2 foods-12-01210-f002:**
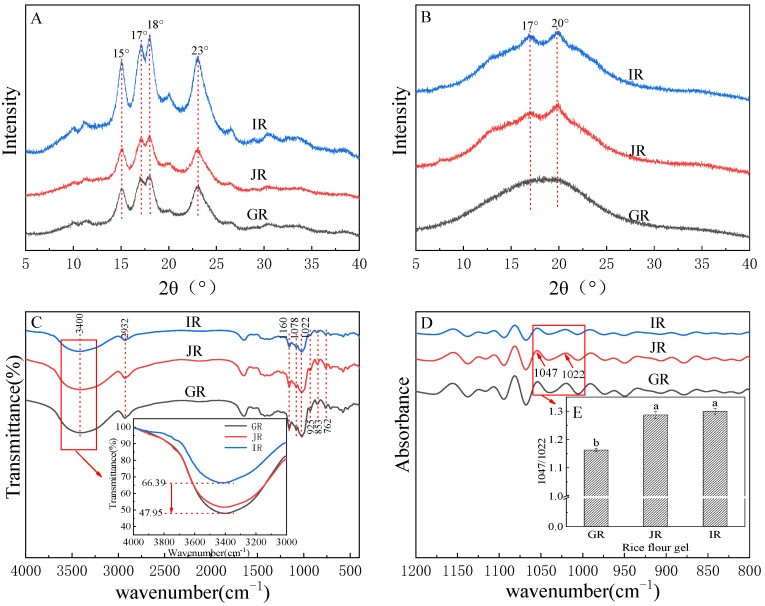
X-ray diffraction patterns of rice flour (**A**); the gel retrograded at 4 °C for 7 days (**B**); the FTIR spectra (**C**) and the deconvolved FTIR spectra (**D**) of the gel retrograded at 4 °C for 7 days; and the relative amount of short-range ordered structure to the amorphous region of the gel retrograded at 4 °C for 7 days (**E**), chart bars with no letter in common are significantly different for *p <* 0.05.

**Figure 3 foods-12-01210-f003:**
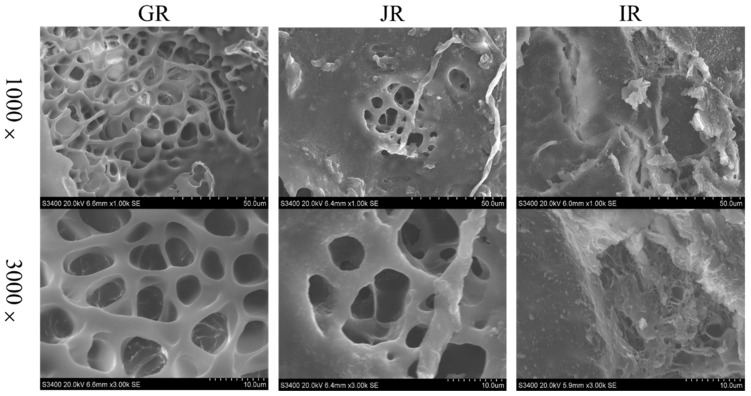
SEM images of the gel retrograded at 4 °C for 7 days.

**Table 1 foods-12-01210-t001:** Thermal properties of rice flour and rheological parameters of gel.

Sample	Thermal Properties					Rheological Parameters		
	To (°C)	Tp (°C)	Tc (°C)	ΔT (°C)	ΔH (J/g)	tan δ (ω = 10 rad/s)	*K**	*n**
GR	64.74 ± 0.48 ^c^	73.35 ± 0.27 ^c^	80.41 ± 0.28 ^b^	15.67 ± 0.74 ^b^	13.25 ± 0.24 ^a^	0.552 ± 0.003 ^a^	11.36 ± 0.55 ^c^	0.3844 ± 0.0029 ^a^
JR	66.15 ± 0.55 ^b^	76.73 ± 0.17 ^b^	83.99 ± 0.62 ^a^	17.84 ± 0.52 ^a^	10.48 ± 0.25 ^b^	0.164 ± 0.004 ^b^	200.3 ± 4.8 ^b^	0.0727 ± 0.0036 ^b^
IR	73.71 ± 0.24 ^a^	77.35 ± 0.04 ^a^	80.60 ± 0.09 ^b^	6.90 ± 0.33 ^c^	7.65 ± 0.05 ^c^	0.088 ± 0.000 ^c^	282.3 ± 5.0 ^a^	0.0589 ± 0.0029 ^c^

Results were expressed as means ± standard deviation (*n* = 3); the values with different letters in the same column are significantly different at *p <* 0.05; To, Tp, and Tc indicate the onset, peak, and conclusion temperatures of the gelatinization process, respectively; ΔT indicates the temperature range of the gelatinization process, and was calculated by Tc-To; ΔH indicates the enthalpy in the gelatinization process; tan δ indicates the loss tangent calculated by dividing G″ by G′; *K** and *n** indicate the consistency coefficient and flow behavior index, respectively; GR indicates the Glutinous rice; JR indicates the Japonica rice; and IR indicates the Indica rice.

**Table 2 foods-12-01210-t002:** Thermal properties, retrogradation degree, and relative crystallinity of the gel (retrograded at 4 °C for 7 days).

Sample	Thermal Properties				RD (%)	Relative Crystallinity (%)		
	T_rO_ (°C)	T_r_p (°C)	T_r_c (°C)	ΔHr (J/g)		B-Type	V-Type	Total
GR	50.92 ± 0.09 ^c^	59.24 ± 0.12 ^c^	64.90 ± 0.15 ^c^	0.30 ± 0.01 ^c^	2.27 ± 0.03 ^c^	ND	ND	2.035 ± 0.022 ^c^
JR	51.94 ± 0.09 ^b^	61.52 ± 0.18 ^b^	72.28 ± 0.13 ^b^	2.24 ± 0.05 ^a^	21.41 ± 0.22 ^b^	2.482 ± 0.030 ^b^	5.555 ± 0.007 ^a^	8.912 ± 0.086 ^b^
IR	54.26 ± 0.11 ^a^	63.21 ± 0.13 ^a^	74.69 ± 0.14 ^a^	1.97 ± 0.01 ^b^	25.79 ± 0.16 ^a^	3.542 ± 0.041 ^a^	5.615 ± 0.069 ^a^	10.67 ± 0.16 ^a^

Results were expressed as means ± standard deviation (*n* = 3); Values with different letters in the same column are significantly different at *p* < 0.05; T_rO_, T_r_p, and T_r_c indicate the onset, peak, and conclusion temperature during the melting process; ΔHr indicates the retrogradation enthalpy; RD indicates the retrogradation degree calculated by dividing the retrogradation enthalpy by gelatinization enthalpy and multiplying it by 100%; and ND indicates not detected.

## Data Availability

The data are available from the corresponding author.

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
