# Peer review of "The Role of Amylose in Gel Forming of Rice Flour"

_foods, 2023, doi:10.3390/foods12061210_

Round 1

Reviewer 1 Report

Manuscript: The role of amylose in gelatinizing process of rice flour

Manuscript Number: foods-2211152

Paper is well structured and all experimental procedures are clearly described. Manuscript contains some interesting observations, however, there are several additional comments regarding this manuscript:

Keywords→ arrange in alphabetical order.

Line 57: (Li et al., 2021) → [1]

Line 65: (Li et al., 2016)  → [4]

The novelty of the work should be emphasized in the introduction section.

Has this material been identified by exsicata?

There was no enough discussion or analysis of the results. The author should explain and clearly discuss this part based on scientific knowledge. It is better to compare the results with more similar recent works. And discuss the superiority of the work.

Line 471, Table 2 Thermal properties, retrogradation degree and relative crystallinity of gel: change title to → Table 2 Thermal properties, retrogradation degree and relative crystallinity of gel (retrograded at 4℃ for 7 days)

Author Response

Reviewer 1

Comment 1:

Keywords → arrange in alphabetical order.

Response:

Thanks for your suggestion, we have rearranged keywords in alphabetical order as “Amylose, Gelatinizing process, Gels, Rice flour”.

Comment 2:

Line 57: (Li et al., 2021) → [1]

Response:

Thanks for your suggestion, the cited references were presented in arabic numeral order in the full text.

Comment 3:

Line 65: (Li et al., 2016) → [4]

Response:

Thanks for your suggestion, the cited references were presented in arabic numeral order in the full text.

Comment 4:

The novelty of the work should be emphasized in the introduction section.

Response:

Thanks for your suggestion, we have added some content in the introduction section to highlight the originality of the manuscript.

Comment 5:

Has this material been identified by exsicata?

Response:

Thanks for your suggestion, the rice samples have been identified.

Comment 6:

There was no enough discussion or analysis of the results. The author should explain and clearly discuss this part based on scientific knowledge. It is better to compare the results with more similar recent works. And discuss the superiority of the work.

Response:

Thanks for your suggestion, we have made further discussion and compared the results with previous study in Results and discussions section, the superiority of the study was highlighted.

Comment 7:

Line 471, Table 2 Thermal properties, retrogradation degree and relative crystallinity of gel: change title to → Table 2 Thermal properties, retrogradation degree and relative crystallinity of gel (retrograded at 4℃ for 7 days)

Response:

Thanks for your suggestion, we have modified the title of the table 2 following the comments as “Thermal properties, retrogradation degree and relative crystallinity of gel (retrograded at 4℃ for 7 days)”. 

Reviewer 2 Report

The manuscript has investigated the role of amylose in gelatinizing process of rice flour. The topic is interesting; however, However, what is the novelty of the current work? It should be mentioned at the end of the introduction part. Because, the role of amylose in starch gelatinization was already evaluated in many studies.

Others:

1. Please check the style of citations.

2. L 61; What is the difference between amylose/amylopectin of GR, IR and IR?

3. L 63-65; Please check the defination of starch gelatinization.

4. To, Tp, and Tc; not To, Tp, and Tc.

5. Section 2.8; Please mention the magnification.

6. Section 3.1; Compare your results with other studies.  

7. L 197; Is there a difference between starch pasting and starch gelatinization?

8. Sections 3.4, 3.5, 3.6, and 3.7; It is better to change "gel" to "retrograded
gel" in the titles.

Author Response

Reviewer 2

Comment 1:

Please check the style of citations.

Response:

Thanks for your suggestion, we have modified the format of references following the “foods” journal style.

Comment 2:

L 61; What is the difference between amylose/amylopectin of GR, IR and IR?

Response:

Thanks for your suggestion, the difference between amylose/amylopectin of GR, IR and IR was inserted in the end of the first paragraph.

Comment 3:

L 63-65; Please check the defination of starch gelatinization.

Response:

Thanks for your suggestion, this sentence describes the process of converting into a gelatinous form. Therefore, the word “gelatinization” was revised as “the process of gel forming”.

Comment 4:

To, Tp, and Tc; not To, Tp, and Tc.

Response:

Thanks for your suggestion, we have modified the section following the comments.

Comment 5:

Section 2.8; Please mention the magnification.

Response:

Thanks for your suggestion, we have added the magnification times in Section 2.8.

Comment 6:

Section 3.1; Compare your results with other studies.

Response:

Thanks for your suggestion, we have added some content to compare the present results with the previous studies in Section 3.1.

Comment 7:

L 197; Is there a difference between starch pasting and starch gelatinization?

Response:

Thanks for your suggestion. In the previous version of the study, “Pasting” indicates the process of converting into a solution form, “Gelatinization” indicates the process of converting into a gelatinous form. Based on published articles, the process that starch converts into a solution form should be described by the word “Gelatinization”, and the process that starch converts into a gelatinous form should be described by the phrase “The process of gel forming”. In the present version, we have revised the expression,

Comment 8:

Sections 3.4, 3.5, 3.6, and 3.7; It is better to change "gel" to "retrograded gel" in the titles.

Response:

Thanks for your suggestion, we have revised the titles of 3.4, 3.5, 3.6, and 3.7 sections following the comments.

Reviewer 3 Report

The paper treats the preparation of gels from three varieties of rice flour namely glutinous rice (GR), Japonica rice (JR) and Indica rice (IR) ‒ with different amylose contents: low, medium and high. The rheological and thermal properties of gels prepared were assessed. It was found the increasing of the strength and elasticity of gels with the increase of the amylose content.

The research is of interest and could be used to further development of rice product as the authors mentioned in Conclusion.

Conclusion part is too general and should be reformulated based on the main findings of the research.

Minor corrections:

Line 61:

   “japonica rice and indica rice” use capital letter :  Japonica rice and Indica rice 

Line 311

 … vibration of C─O in anhydrous glucose unit [26], In these spectra, the absorption peak         

The point is missing at the end of the sentence.

Fig. 2. D, please verify the code of blue line spectrum, that is probably IR not JR

Author Response

Comment 1:

Line 61: “japonica rice and indica rice” use capital letter : Japonica rice and Indica rice

Response:

Thanks for your suggestion, we have changed the “glutinous rice, japonica rice and indica rice” into “Glutinous rice, Japonica rice and Indica rice”.

Comment 2:

Line 311

 … vibration of C─O in anhydrous glucose unit [26], In these spectra, the absorption peak. The point is missing at the end of the sentence.

Response:

Thanks for your suggestion, we have added the key-point in the end of the sentence.

评论3:

无花果。2. D,请验证蓝线光谱的代码,那可能是IR而不是JR

响应:

感谢您的建议,我们已将图中的蓝线频谱代码JR更改为IR。2.D,我们也检查了整个稿件中的类似问题。

Round 2

Reviewer 1 Report

No Comment

Reviewer 2 Report

The manuscript is now acceptable.